# Clinical and Genetic Aspects of Phelan–McDermid Syndrome: An Interdisciplinary Approach to Management

**DOI:** 10.3390/genes13030504

**Published:** 2022-03-12

**Authors:** Francisco Cammarata-Scalisi, Michele Callea, Diego Martinelli, Colin Eric Willoughby, Antonio Cárdenas Tadich, Maykol Araya Castillo, María Angelina Lacruz-Rengel, Marco Medina, Piercesare Grimaldi, Enrico Bertini, Julián Nevado

**Affiliations:** 1Pediatric Service, Regional of Antofagasta Hospital, Antofagasta 1240835, Chile; francocammarata19@gmail.com (F.C.-S.); dr_cardenas2000@yahoo.es (A.C.T.); marcomedina-0@hotmail.com (M.M.); 2Pediatric Dentistry and Special Dental Care Unit, Meyer Children’s University Hospital, 50139 Florence, Italy; 3Unit of Metabolism, Bambino Gesù Children’s Research Hospital IRCCS, 00165 Rome, Italy; 4Genomic Medicine, Biomedical Sciences Research Institute, Ulster University, Coleraine Campus, Coleraine BT52 1SA, Northern Ireland, UK; c.willoughby@ulster.ac.uk; 5Clinical Laboratory, Regional of Antofagasta Hospital, Antofagasta 1240835, Chile; maykol.araya@redsalud.gov.cl; 6Neuropediatrics Service, Childcare and Pediatric Department, University of Los Andes, Mérida 51101, Venezuela; lacruz_rengel@hotmail.com; 7Department of Public Health and Pediatric Sciences, University of Torino, 10126 Torino, Italy; piercesare.grimaldi@unito.it; 8Unit of Neuromuscular and Neurodegenerative Disorders, Laboratory of Molecular Medicine, Department of Neurosciences, Bambino Gesu’ Children’s Research Hospital IRCCS, 00165 Rome, Italy; enricosilvio.bertini@opbg.net; 9Instituto de Genética Médica y Molecular (INGEMM), Instituto de Investigación del Hospital Universitario La Paz (IdIPaz), 28046 Madrid, Spain; jnevadobl@gmail.com or; 10Centro de Investigación Biomédica en RED de Enfermedades Raras (CIBERER), 28046 Madrid, Spain; 11ITHACA, European Reference Network on Rare Congenital Malformations and Rare Intellectual Disability, Hospital La Paz, 28046 Madrid, Spain

**Keywords:** Phelan–McDermid syndrome, *SHANK3*, etiology, evaluation, treatment

## Abstract

Phelan–McDermid syndrome (PMS) is a rare, heterogeneous, and complex neurodevelopmental disorder. It is generally caused by a heterozygous microdeletion of contiguous genes located in the distal portion of the long arm of chromosome 22, including the *SHANK3* gene. Sequence variants of *SHANK3*, including frameshift, nonsense mutations, small indels and splice site mutations also result in PMS. Furthermore, haploinsufficiency in *SHANK3* has been suggested as the main cause of PMS. *SHANK3* is also associated with intellectual disability, autism spectrum disorder and schizophrenia. The phenotype of PMS is variable, and lacks a distinctive phenotypic characteristic, so the clinical diagnosis should be confirmed by genetic analysis. PMS is a multi-system disorder, and clinical care must encompass various specialties and therapists. The role of risperidone, intranasal insulin, insulin growth factor 1, and oxytocin as potential therapeutic options in PMS will be discussed in this review. The diagnosis of PMS is important to provide an appropriate clinical evaluation, treatment, and genetic counseling.

## 1. Introduction

Phelan–McDermid syndrome (PMS) is a rare, heterogeneous, and complex disorder of neurological development [1,2,3,4,5,6,7,8]. The principal cause of PMS is a heterozygous microdeletion of contiguous genes (including *SHANK3*, MIM *606230, or just part of this gene), located in the distal portion of the long arm of chromosome 22 [6,8,9,10], of which 70% occur on the paternal chromosome [11]. Other cases result from single nucleotide variants in the SH3 and Multiple Ankyrin Repeat Domains 3 gene (*SHANK3*) [2,3,6,8,10,12,13,14]. This gene was initially associated with a 4.5-year-old boy manifesting all the features of a 22q13.3 deletion syndrome. Bonaglia and colleagues found in the patient’s karyotype a de novo balanced translocation between chromosomes 12 and 22, with the breakpoint in the 22q13.3 critical region of the 22q distal deletion syndrome [15]. Furthermore, variants in *SHANK3* are also associated with intellectual disability in 2% [6,10,14], autism spectrum disorder in 0.5–2%, and schizophrenia in 0.6–2.16% of cases [6,10,14]. *SHANK3* is therefore an attractive target for pharmacological intervention in the treatment of these neurodevelopmental disorders with neurological and neuropsychiatric manifestations [6,10,14].

Watt et al. [16] first described PMS in 1985, in a 14-year-old male adolescent with severe intellectual disability, absence of language, minor dysmorphia, and normal muscle tone. In 1988, Herman and colleagues identified a terminal deletion at 22q13.3 in PMS [17], and Phelan and McDermid described the hypotonia associated with the de novo deletion in 22q13.3 in a newborn who later presented with global developmental delay, normal growth, and minor facial dysmorphias [18].

The occurrence rate has been estimated in the range of 2.5–10 per million births, although this is likely to be underestimated (National Organization for Rare Disorders. Phelan–McDermid syndrome), and is genetically detectable by array CGH for a small-size deletion in 22q13.3, or the need for direct or NGS sequencing of *SHANK3* to support the specific identification of patients [10,14]. According to the Phelan–McDermid Syndrome International Registry (https://www.pmsf.org/registry/, accessed on 10 November 2021), at least 1200 cases have been reported worldwide [12].

## 2. Clinical Features

The PMS phenotype can vary widely and lacks a distinctive characteristic phenotype [6]. Clinically, it presents with neonatal hypotonia [2,6,10,19], global delay of development [2,3,8,14,19], absence or severe alteration of language [3,4,6,7,10,14,20], highly variable levels of cognitive functioning with moderate-to-severe intellectual disability in 77% [3,4,5,6,7,8,10,19,20], behavioral disturbances: attention deficit hyperactivity disorder in 36% [2,3,7,8,10], autism spectrum disorder in 84% [4,6,8,14,20], sleep disturbances, stereotyped repetitive movements [5,19], reduced sensitivity to pain, an inability to regulate sweating [8], and seizures of different types [2,3,7,19,20]. Seizures are mostly febrile and do not require medication; however, grand mal seizures, focal seizures, and absence seizures have been reported. No characteristic EEG findings have been described in PMS.

Neuroradiological (for example BRAIN IMAGE) studies in PMS have documented reduced myelination, frontal lobe hypoplasia, corpus callosum thinning or agenesia, ventriculomegaly, and focal cortical atrophy [21].

PET studies in eight children with PMS showed a localized dysfunction of the left temporal polar lobe and significant hypoperfusion of the amygdala compared to 13 children with idiopathic intellectual disability [22].

Growth is normal [5,10], and dysmorphic features are generally mild and include: dolichocephaly [3,5,6,7,8,10,14], long eyelashes [10], midfacial hypoplasia [7], prominent cheeks [10], bulbous nasal tip, pointed chin [10], alterations in ears, fleshy hands, and dysplastic toenails [23]. Additional features include endocrine, immune, ocular, cardiopulmonary, gastrointestinal, renal malformations, and lymphedema abnormalities (see, Table 1), [1,2,3,6,7,8,10,14,19,20,23].

### Natural History

The least-affected clinical skill is motor functioning, with gross motor function being stronger than fine motor function [8,24]. The regression in motor and social skills with a slow and possibly incomplete recovery may occur after acute paroxysmal events [18,24]. The intellectual disability is less striking in younger children than in older children, as relative developmental functioning decreases with increasing age. The most affected developmental skill is language, although receptive language is usually reported to be stronger than expressive language, and social skills essentially present with symptoms in the autism spectrum [18,19].

Most children with PMS seem to acquire basic skills such as walking, but subsequent extension and refinement of these skills is generally very poor. These deficiencies are less striking in younger than older children, a phenomenon known as ‘growing into deficit’. In addition, deficits in adaptive behavior further hamper cognitive development. No improvement of developmental milestones is a common finding in this population, and is important to take into account when evaluating development and treatment effects. At the level of psychopathology, typical characteristics are related to the affective and anxiety domains, with, for example, impulsive, irritable, and demanding behaviors. In post-adolescent patients, psychiatric symptoms such as atypical bipolar disorder appear to be more prominent [19]. This entity is still under-diagnosed in adults who may present with deterioration in cognitive functioning and atypical bipolar disorders with loss of acquired skills [8].

## 3. Etiopathogenesis

The genetic landscape of PMS is broad and the mechanism in which the deletion occurs can be caused by a variety of rearrangements, including terminal or interstitial deletions, unbalanced translocations, ring chromosomes, and other more complex alterations [3,6,14,25] (Figure 1). Terminal and de novo deletions contribute to the majority of chromosomal anomalies and are reported in approximately 80% of cases. The deletion size can vary from less than 100 kb to 9.2 Mb (4.5 Mb on average), resulting in the loss of the single *SHANK3* gene, and ranges from 22q13.2 to 22q13.31-3 [3,14]. Moreover, PMS commonly involves the loss of other genes, such as *PARVB* (MIM *608121) [8] and *SULT4A1* (MIM *608359), and this have been suggested to contribute to phenotype severity [8,26]. Unbalanced translocations or a ring chromosome involving chromosome 22 represent the remaining 20% of cases [3,6,10,14], of which approximately 50% are inherited interchangeably from a balanced carrier parent [9]. Some families with affected siblings have been described, so the origin of the rearrangement may be due to germinal mosaicism in one of the parents [27,28,29].

Interestingly, PMS patients with ring chromosome 22 may develop neurofibromatosis type 2 (NF2) [13], caused by pathogenic variants in *NF2*, as this gene is located at 22q12.2, adjacent to the PMS deletion region. The risk for NF2 is due to the instability of ring chromosomes during mitosis and follows a two-hit model. The first hit is the loss of *NF2* during mitosis; the second hit is a somatic mutation of the remaining *NF2* allele [30]. Out of 44 individuals evaluated at the Greenwood Genetic Center or by international collaborators or identified through the PMS International Registry with a ring chromosome 22, 7 (16%) carried a diagnosis of NF2, therefore, this condition is not uncommon [31].

Children with ring chromosome 22 should be monitored for NF2 signs, with baseline and annual ocular, dermal, and neurologic examinations between ages 2 and 10 years with annual audiology screening and brain MRI every two years after age 10 years [32].

SHANKproteins are major scaffolds of the postsynaptic density of excitatory synapses, linking neurotransmitter receptors and ion channels to the actin cytoskeleton and G-protein-coupled signaling pathways, and mutations in *SHANK3* gene are associated with autism and intellectual disability [33], as well as psychiatric diseases, such as schizophrenia and bipolar disorder [7]. SHANK3 protein has a critical function in synaptic plasticity by modulating dendrite formation, playing an important role in the development of synapses [3,7,10,14] and the maintenance of long-term potentiation [7]. Defects in SHANK3 protein can also cause a reduction in activated hyperpolarization of cation channels [10,14], which may explain some of the observed phenotypic characteristics in patients [10].

Genotype–phenotype studies are made preferentially in cases with deletions and correlate positively with the number and/or severity of some of the clinical manifestations [3,8], some neurological manifestations including hypotonia [7,34], delayed motor development, deficiencies in social communication related to autism spectrum disorders, aggressive behavior, and intellectual disability [23]. Therefore, individuals with small terminal deletions or with sequence variants in *SHANK3* gene may have a less severe phenotype. Those cases with major deletions or truncated nonsense mutations in *SHANK3* have been associated with autism spectrum disorders and intellectual disability [3]. In a mouse model of autism spectrum disorders, a truncated mutant form of *Shank3* displayed altered plasticity, anxiety-like, motor, social, communications and stereotyped behaviors [35,36,37], which were attributable to reduced synaptic plasticity in the hippocampal–medial prefrontal cortex pathway [37]. Individuals with PMS can exhibit sensitivity to stress, resulting in behavioral deterioration. Using a mouse model, swim stress produces an altered transcriptomic response in pyramidal neurons that impacts genes and pathways involved in synaptic function, signaling, and protein turnover. Several lines of evidence demonstrate that *Shank3* expression is regulated by Homer protein homolog 1a (Homer1a), which is part of the Shank3-mGluR-N-methyl-D-aspartate (NMDA) receptor complex and is super-induced and implicated in the stress response. The interaction between stress and genetics and focus attention on activity-dependent changes may contribute to pathogenesis [38].

However, the relationship is not consistent, since patients with deletions of similar size can have variable phenotypes [7]. Individuals with infrequent interstitial deletions, without involvement of the *SHANK3* gene, may exhibit an indistinguishable phenotype, raising the possibility that many of the characteristics of PMS may be under the influence of other additional genes located in the 22q13.32 [14] region, or may affect regulatory regions of the *SHANK3* gene or have a position effect [2,4].

A deletion that has been identified in PMS contains a region that houses three genes: *ACR*, (MIM *102480), *RABL2B* (MIM *605413) and *SHANK3*, the latter being the strongest candidate for neurobehavioral symptoms [39]. Other genes can potentially influence neurological abnormalities in PMS, such as the *MAPK8IP2* gene (OMIM *607755), located approximately 70 Kb proximal to *SHANK3*, but often deleted together. *MAPK8IP2* is highly expressed in the brain at the level of the posterior synaptic space, and studies in mice show that the absence of the MAPK8IP2 protein results in an abnormal dendritic morphology and cognitive and motor deficits. However, it is not yet clear how deletions in *SHANK3* and *MAPK8IP2* specifically contribute to PMS [7]. The mentioned *SULT4A1* gene encodes a cytosolic sulfotransferase highly expressed at postsynaptic sites, which modulates neuronal branching complexity and dendritic spines’ formation, negatively regulates the catalytic activity of Pin1 toward PSD-95, and facilitates NMDAR synaptic expression and function [26].

The genetic heterogeneity of PMS underscores the importance of studying a wide range of alterations [3]. Various case series have been published that have investigated the correlation of the clinical characteristics [3,14], with a wide heterogeneity in the expression and severity of the phenotype [3].

## 4. Molecular Diagnosis

The diagnosis of PMS is based on confirmed chromosomal and/or molecular analysis. The karyotype is able to identify large deletions and other molecular techniques should be used for the genetic characterization of the deletion (FISH, MLPA, array CGH/SNPs). In the case of variants in *SHANK3*, either conventional Sanger or next-generation sequencing (NGS) through targeted capture, or whole-exome sequencing (WES), must be performed to sequence the *SHANK3* gene. The advantages and disadvantages of the different and main experimental approaches in the diagnosis of PMS are highlighted in Figure 1 and Table 2.

The most reasonable sequence or diagnostic algorithm for deletions is CMA (chromosomal microarray). It is necessary to identify the size of the deletion (and the genes deleted), if the deletion is terminal or interstitial, and if there are additional rearrangements. In terminal deletions, a karyotype or FISH will be required to identify if it is the consequence of a translocation or the consequence of a ring. Prenatal diagnosis is possible and is based on cytogenetic and molecular analysis after an invasive amniocentesis test or chorial biopsy. Recently, extended non-invasive prenatal diagnosis has been able to establish variations in copy number in the region affecting PMS and other syndromes as a prenatal screening. However, diagnostic confirmation is still required by invasive techniques, and the application of microarrays CMA, FISH, MLPA, or karyotype could be necessary.

## 5. Differential Clinical Diagnosis

The differential diagnosis of PMS (Table 3) can be challenging, given the clinical variability in phenotype, and includes: autism spectrum disorders, cerebral palsy, Prader-Willi syndrome (MIM #176270), Angelman syndrome (MIM #105830), velocardiofacial syndrome (MIM #192430), fragile X syndrome (MIM #300624), FG (MIM #305450), Williams–Beuren syndrome (MIM #194050), and Smith–Magenis syndrome (MIM #182290) [12,13,23]. Furthermore, the trichorinophalangeal syndromes (MIM #190350), Clark–Baraitser (MIM #300602) and Sotos (MIM #117550) are also considered differential diagnoses [5].

## 6. Interdisciplinary Clinical Management

The care team should be interdisciplinary and should include neonatologists, pediatricians, neurologists, psychiatrists, endocrinologists, immunologists, cardiologists, gastroenterologists, nephrologists, dentists, as well as speech, respiratory, physical, and occupational therapists who provide follow-up evaluations, exercises and infant massage, and lactation consultants, experts in the evaluation of oral feeding strategies and social workers. All these clinical staff should be coordinated or supervised by a geneticist or a neuropsychiatrist who will provide clinical follow-up and genetic counseling to the family.

More detailed information on developmental characteristics in children is clearly necessary for care and would contribute to improving counselling of parents, identification of specific problems, adequate and individualized support for the persons with PMS [38].

The first line of treatment must address the areas of motor, communication, and language development. Early intervention programs of intensive physical and occupational therapies are highly recommended for motor skills deficits. While evaluating any deterioration of developmental skills, frequent monitoring should be performed to detect and manage medical comorbidities [12,40]. Subsequently, the areas that should be covered are those related to cognitive and behavioral characteristics. These areas should be evaluated and followed to adapt supportive and therapeutic strategies to meet individual needs [8].

An ophthalmological evaluation must be performed to monitor visual development and identify any refractive error or strabismus. Myopia, strabismus, and retinitis pigmentosa have been reported. Dental alterations, such as malocclusion, are frequent and can be serious in some cases, for which reason orthodontics or surgical correction can reduce the risk of dental caries, periodontal disease and relieve pressure on the temporomandibular joint [41].

Patients with PMS can be prone to immune system dysfunction, resulting in recurrent ear and upper respiratory tract infections (complicated by low muscle tone, airway abnormalities, and sputum clearance), asthma, and seasonal and food allergies. Cases of autoimmune hepatitis, atopic dermatitis and recurrent staphylococcal skin infections/cellulitis have been described. Lymphedema has been found in 24% of cases. Clinical management is mainly supportive, but as the underlying pathways responsible for deficits are more clearly understood, there will be the potential to develop more targeted therapies in the future [12].

## 7. Therapeutic Options

The long-term plan for managing PMS patients should be continuous and interdisciplinary, with support to maximize developmental outcome. Early intervention services should be provided to the family in the first three years. However, life expectancy is uncertain and depends on each individual case [42].

Several pharmacological strategies have been assayed. The use of risperidone (1 mg/day) produced significant improvements in behavioral disturbances in PMS on the global clinical impression scale. It is hypothesized that by blocking dopamine 2 receptors, NMDA transmission is promoted, and glutamatergic dysregulation caused by the loss of SHANK3 is reversed [12].

Intranasal insulin was studied in six PMS patients and resulted in marked short-term improvements in gross and fine motor activities, cognitive functions and educational level. Similarly, long-term positive effects were found for gross and fine motor activities, nonverbal communication, cognitive functions, and autonomy. However, one patient showed changes in balance, extreme sensitivity to touch and general loss of interest, and one patient complained of intermittent nasal bleeding [43]. No adverse effects on glucose levels or HbA1c levels were found [12].

To validate this effect, a randomized, double-blind, placebo-controlled clinical trial was conducted in 25 patients aged 1 to 16 years with a molecularly confirmed 22q13.3 deletion involving the *SHANK3* gene. A significant effect was found for cognition and social skills for children older than 3 years, who generally show a decrease in development. Intranasal insulin did not cause serious adverse events. However, clinical trials in larger study populations are required to test the therapeutic effect and safety in PMS [39].

Insulin growth factor 1 (IGF-1) is a small polypeptide that crosses the blood–brain barrier and occurs in higher concentrations during development of the central nervous system, which promotes neuronal maturation and synapse formation. Studies with human neurons created from pluripotential stem cells induced from PMS patients showed that excitatory synapse transmissions can be reversible when treated with IGF-1 in vitro and with overexpression of the SHANK3 protein. IGF-1 also resulted in improvement of NMDA- and AMPA-mediated responses, reducing the decay rate of the NMDA receptor. A control study shows that the administration of IGF-1 in children with PMS improved autistic behavior, probably due to its effect on synapse development and plasticity. This treatment restored the normal density of the dendritic spine in neurons. IGF-1 was also found to work by increasing protein kinase B levels, and therefore direct treatment is another option. Preliminary results from the first clinical study of IGF-1 administration in PMS patients suggest tolerability and significant improvements in both social impairments and restrictive behaviors. No serious adverse effects have been documented and it was determined to be safe in this population. This study is currently in phase 2 and has been expanded to include a larger sample. The cost and availability of the drug are currently restrictive [44].

The acute administration of intra-cerebroventricular oxytocin reversed the deficits of synaptic plasticity in vitro and in vivo in a Shank3 knockout mouse [40]. This was the first study to report that oxytocin may not only reverse this deficit, but also improve behavioral plasticity, suggesting that a reversal effect on synaptic plasticity, specifically long-term potentiation, may be the basis of the improvement in behavior. However, further studies are needed to determine the effect of *SHANK3* mutations on the oxytocin system to understand whether the disturbance could explain some of the observed behavioral phenotypes and changes related to plasticity [45]. The variable genetic mechanisms contributing to PMS will impact the search for therapeutic interventions [6].

Using induced pluripotent stem cell technology and transcriptomic studies, some authors have shown in PMS-derived hiPSC neurons have an impact in several developmental pathways, such as altered pre- and post-synaptic signaling and Wnt and ECM signaling [37]. Active compounds were evaluated for efficacy in correcting dysfunctional networks of neurons differentiated from individuals with deleterious point mutations in *SHANK3*. Among 202 compounds tested, lithium and valproic acid showed the best efficacy in correcting *SHANK3* haploinsufficiency-associated phenotypes in cells. Lithium pharmacotherapy was subsequently provided to one patient, and after one year, an encouraging decrease in autism severity was observed [46].

The pharmacological augmentation of mGluR5 activity using 3-cyano-N-(1,3-diphenyl-1H-pyrazol-5-yl)-benzamide as the positive allosteric modulator of these receptors restored mGluR5-dependent signaling (DHPG-induced phosphorylation of ERK1/2) and normalized the frequency of mEPSCs in Shank3 knockdown neurons. These data demonstrate that a deficit in mGluR5-mediated intracellular signaling in Shank3 knockdown neurons can be compensated by 3-cyano-N-(1,3-diphenyl-1H-pyrazol-5-yl)-benzamide; this raises the possibility that pharmacological augmentation of mGluR5 activity represents a possible new therapeutic approach for patients with Shank3 mutations [47,48].

Additionally, treatment strategies have used lithium valproic acid and/or quetiapine, which appears to be a rational pharmacological choice for subjects with atypical bipolar disorder who require a mood-stabilizing pharmacotherapy [49]. Secondly, techniques from the neuropsychological and cognitive behavioral domain, such as goal management and perspective taking training, should be applied to compensate for the impaired orchestration of cognition and emotion. Such a combined treatment strategy is meant to reduce the risk for relapse of a bipolar episode and to reduce hyper-reactivity [19].

Interestingly, the treatment of behavioral abnormalities in PMS patients is complicated by pharmacogenomics issues, as the CYP2D6 enzyme, which metabolizes antidepressants and antipsychotics and is encoded by *CYP2D6* gene, which maps to 22q13.2 and is lost in subjects with deletions larger than 8 Mb [50].

For PMS patients with autism spectrum disorders carrying *SHANK3* mutations, a rational approach could also be the use of histone deacetylase (HDAC) inhibitors. In fact, a short treatment with a class I HDAC inhibitor such romidepsin has been demonstrated to alleviate social deficits in Shank3-deficient mice, which persisted for ~3 weeks [51].

Further, treating of Shank3-deficient mice with a 4-week ketogenic diet, which can act as an endogenous inhibitor of class I HDAC via the major product β-hydroxybutyrate, elevates the level of histone acetylation in prefrontal cortex neurons. Ketogenic diet treatment lead to the prolonged rescue of social preference deficits in Shank3-deficient mice, elevated the transcription and histone acetylation of Grin2a and Grin2b and restored the diminished N-methyl-D-aspartate (NMDA) receptor synaptic function in prefrontal cortex neurons [52]. HDAC2 transcription was upregulated in these mice, and knockdown of HDAC2 in the prefrontal cortex rescued their social deficits, highlighting an epigenetic mechanism underlying social deficits linked to Shank3 deficiency [51]. The role of valproic acid as a HDAC inhibitor could explain this efficacy in the treatment of some of the symptoms in patients with *SHANK3* mutations, as reported in Fragile X syndrome [53].

## 8. Conclusions

PMS is a complex, heterogeneous, and underdiagnosed entity. Recognizing it is important in order to provide an appropriate evaluation and appropriate treatment option. There are various diagnostic techniques according to the etiological cause. *SHANK3* appears to be the critical gene responsible for the PMS phenotype when a heterozygous rearrangement or deletion is detected within the region 22q13.2 to 22q13.31-3, although other genes may contribute to severity or modulate its phenotypic presentation. Preliminary investigations should be focused on the genotype–phenotype correlation, at least in microdeletion cases. A complete medical history and a thorough physical examination are the first step in the evaluation. In addition, an interdisciplinary and individualized therapeutic follow-up is required, as well as imparting adequate genetic counseling according to the etiological cause. The pediatrician must become familiar in order to play a fundamental role in the coordination of care through the different subspecialties involved in this care, such as family support, educators, and therapists.

## Figures and Tables

**Figure 1 genes-13-00504-f001:**
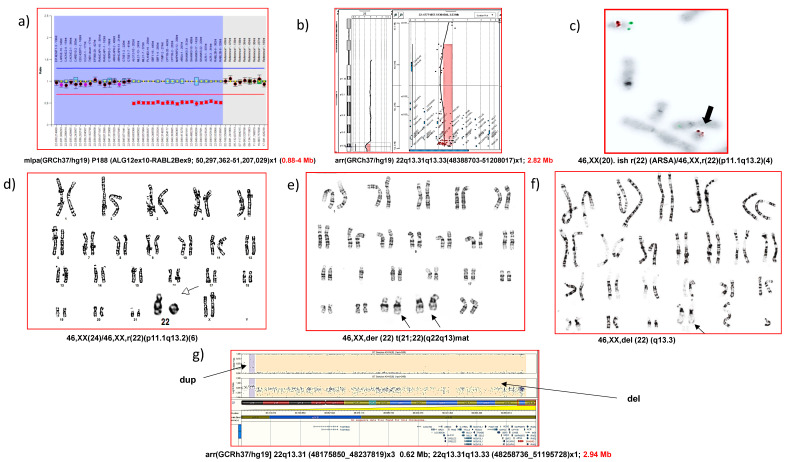
Different genetic rearrangements present in PMS, established by different cytogenetic/molecular techniques: (**a**) interstitial deletion by MLPA; (**b**) terminal deletion by microarray CGH; (**c**) terminal deletion and ring chromosome by FISH; (**d**) terminal deletion and ring chromosome by karyotype (GTL bands), (**e**) unbalanced translocation by karyotype, (**f**) terminal deletion by karyotype, (**g**) terminal deletion and other adjacent SNP-array rearrangements.

**Table 1 genes-13-00504-t001:** Clinical findings associated with PMS [1,2,6,7,8,10,14,19,20,23].

Findings	Percentage (%)	Findings	Percentage (%)
Hyperextensibility	86	Dimple in sacrum	25
Hypotonia	73	Lymphedema	25
Bulbous nasal tip	65	Febrile/nonfebrile seizure	24
Alterations in ears	63	Hypertelorism	22
Long eyelashes	58	Kyphoscoliosis	22
Large, fleshy hands	53	Sunken eyes	19
Epicanthal folds	48	Malocclusion/separated teeth	19
Periorbital fullness	46	Macrocephaly	17
Dysplastic/hypoplastic nails	45	Esotropia/strabismus	17
Pointed chin	44	Sparse/spiral hair	16
Sleep disturbance	44	Wide nasal bridge	16
Gastroesophageal reflux	43	Long philtrum	16
Increased tolerance to pain	42	Microcephaly	12
Constipation/diarrhea	40	Micrognathia	12
Dolichocephaly	37	clinodactyly of the fifth finger	12
High narrow palate	36	Short stature/growth retardation	12
Syndactyly of 2nd and 3rd toes	34	Tall stature/accelerated growth	11
Brain abnormalities (imaging)	32	Malar hypoplasia	9
Prominent lips	31	Congenital heart disease	8
Recurrent respiratory infections	30	Precocious or delayed puberty	6
Eyelid ptosis	29	Low-set ears	5
Kidney disorders	27	Hypothyroidism	5
Prominent cheeks	25	Midface hypoplasia	3

**Table 2 genes-13-00504-t002:** List of advantages and disadvantages of laboratory techniques in the diagnosis of PMS.

	Large Deletions	Cryptic Deletions	Balanced Translocations	Size	Mosaics	UPD	Other Regions
Karyotype	+	−	+	−	+	+	+ Large
FISH	+	+	+	−	+	+	−
MLPA	+	+	−	+ Partial	−	+	−
aCGH	+	+	− Yes, in imbalance	+	+	+	+
SNParray	+	+	− Yes, in imbalance	+	+	−	+

+ indicates “able to detect”; − indicates “not able to detect”.

**Table 3 genes-13-00504-t003:** Differential diagnosis of PMS.

Findings/Entities	PMS	Autism	Paralysis Cerebral	Prader- Willi	Angelman	Velocardiofacial	Fragile X	FG	Williams	Smith- Magenis
Subtle dysmorphisms	+	−	+	+	+	+	+	− *	+ **	+
Language disturbance	+	+	+		+	+	+	+	−	+
Alteration in socialization	+	+	−	−	+	−	+	−	−	−
Repetitive movements	+	+	−	−	+	−	+	+	−	−
Hypotonia	+	−	+	+	+	+	−	+	+	+
Global developmental delay	+	−	+	+	+	+	+	+	+	+
Feeding difficulties	+	−	+	+	−	+	−	−	+	−
Poor coordination	+	−	+	−	+	−	+	−	+	−

* Present facial dysmorphias are not similar to PMS, ** Characteristics of Williams syndrome.

## Data Availability

Support with the literature review.

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
