# Peer review of "Clinical and Genetic Aspects of Phelan–McDermid Syndrome: An Interdisciplinary Approach to Management"

_genes, 2022, doi:10.3390/genes13030504_

Round 1

Reviewer 1 Report

This is an interesting and comprehensive review on the genetics and clinical features of Phelan-McDermid syndrome. It covers topics ranging from clinical features, diagnosis, management, to therapeutic options. It is a clearly written and easy-to-read review. I have only a few minor suggestions –

  1. The authors mentioned the preclinical genotype-phenotype studies (line 155). This is a fast-moving and important field including models from rodents to primates. Findings from these models uncovers mechanisms underlying behavioral alterations so I think the authors should include and cite more up-to-date findings, e.g., how Shank3 deficiency links with behavior alterations that manifest clinically. They should also include more detailed discussions.
  2. The authors mentioned about using HDAC inhibitor to alleviate symptoms of PMS (line 306). The transcriptomic alteration in PMS models, in vitro as well as in vivo, is very mild under basal conditions (PMID: 34788607, PMID: 32560742, PMID: 31489248) but becomes drastic by mild stress (PMID: 34788607). The authors may want to highlight these findings to rationalize manipulating gene expression as the basis for treating PMS.
  3. The authors mentioned about gene therapy as a potential future option (line 311) but with no discussion. It would be nice if the authors can give some review on the current progress on gene therapy and how these advances can help PMS.

Author Response

REVIEWER 1

Comments and Suggestions for Authors

This is an interesting and comprehensive review on the genetics and clinical features of Phelan-McDermid syndrome. It covers topics ranging from clinical features, diagnosis, management, to therapeutic options. It is a clearly written and easy-to-read review. I have only a few minor suggestions –

  1. The authors mentioned the preclinical genotype-phenotype studies (line 155). This is a fast-moving and important field including models from rodents to primates. Findings from these models uncovers mechanisms underlying behavioral alterations so I think the authors should include and cite more up-to-date findings, e.g., how Shank3 deficiency links with behavior alterations that manifest clinically. They should also include more detailed discussions.

We thank the reviewer for his  suggestion and we included recent evidences in animal models of the role of Shank3 deficiency

  1. The authors mentioned about using HDAC inhibitor to alleviate symptoms of PMS (line 306). The transcriptomic alteration in PMS models, in vitro as well as in vivo, is very mild under basal conditions (PMID: 34788607, PMID: 32560742, PMID: 31489248) but becomes drastic by mild stress (PMID: 34788607). The authors may want to highlight these findings to rationalize manipulating gene expression as the basis for treating PMS.

We thank the reviewer for his  suggestion and we included in the discussion some data about the role  HDAC inhibitors in PMS

  1. The authors mentioned about gene therapy as a potential future option (line 311) but with no discussion. It would be nice if the authors can give some review on the current progress on gene therapy and how these advances can help PMS.

As suggested by the reviewer,we discussed the role of gene therapy approaches and induced pluripotent stem cell technology in PMS

Reviewer 2 Report

In this review Cammarata-Scalisi et al. summarize some of the specific clinical and genetic alterations observed in the Phelan-McDermid syndrome. The review is in general interesting but in several points the review needs to be improved. There are several and important papers related to the Shank3 function and PMS that were not cited and several citations are totally wrong. Clearly, the authors are not very experts on PMS and reported the literature with several mistakes.

Line 62: The paper that first associated PMS with SHANK3 deletion was not cited, Bonaglia et al. Am J Hum Genet. 2001.

Table 2, citations associated with this table are totally wrong, especially the citation n. 3 that describe the characterization of a Shank3 KO mice and not a clinical characterization of the PMS patients. Please provide a correct and complete source of the data reported in table 2.

Line 126: Interstitial deletions in the 22q13 region that cause PMS syndrome were not mentioned, see Wilson et al. Eur J Hum Genet. 2008 and Disciglio et al. Am J Med Genet A. 2014.

Line 139: “Interestingly, PMS patients with ring chromosome 22 may develop [13] (NF2)”, NF2 was not defined.

Lines 148-154: The description of the function of Shank3 is completely wrong, these sentences need to be fully re-written and the citations completely changed. I suggest to the authors carefully read the relative and important references.

Line 171: Other genes besides MAPK8IP2 might contribute to PMS such as SULT4A1 which function has been studied in rodent neurons (Culotta et al. JNeurosci 2019).

Lines 270-273. The citations related to the effect of IGF1 on human neurons are wrongs, the correct one is Shcheglovitov et al. Nature 2013.

Line 294: Litium was also proposed as a potential drug to treat PMS patients with a specific mechanism described in Darville et al EBioMedicine, 2016. This important finding was not cited.

Line 306: Finally other potential therapies with mGlu5 PAM were not cited, Verpelli et al JCB 2011, Vicidomini et al Mol Psychiatry 2017 and Wang et al. Nat Commun 2016.

In general, the authors should carefully check the cited literature in support of the written text. Several papers were cited in the wrong context, this couldn’t be accepted for a review.  

Author Response

REVIEWER 2

Comments and Suggestions for Authors

In this review Cammarata-Scalisi et al. summarize some of the specific clinical and genetic alterations observed in the Phelan-McDermid syndrome. The review is in general interesting but in several points the review needs to be improved. There are several and important papers related to the Shank3 function and PMS that were not cited and several citations are totally wrong. Clearly, the authors are not very experts on PMS and reported the literature with several mistakes.

-First of all, we thank the reviewer for his careful and thorough analysis of our paper, which allowed us to improve it extensively

Line 62: The paper that first associated PMS with SHANK3 deletion was not cited, Bonaglia et al. Am J Hum Genet. 2001.

-We thank the reviewer for his observation, and we included the suggested paper (line 54)

Table 2, citations associated with this table are totally wrong, especially the citation n. 3 that describe the characterization of a Shank3 KO mice and not a clinical characterization of the PMS patients. Please provide a correct and complete source of the data reported in table 2.

-We corrected the citations associated to Table 1( not 2 as stated by the reviewer)

Line 126: Interstitial deletions in the 22q13 region that cause PMS syndrome were not mentioned, see Wilson et al. Eur J Hum Genet. 2008 and Disciglio et al. Am J Med Genet A. 2014.

-We included Wilson et al. Eur J Hum Genet. 2008 as suggested

Line 139: “Interestingly, PMS patients with ring chromosome 22 may develop [13] (NF2)”, NF2 was not defined.

-We corrected the text accodingly

Lines 148-154: The description of the function of Shank3 is completely wrong, these sentences need to be fully re-written and the citations completely changed. I suggest to the authors carefully read the relative and important references.

-We corrected the text accodingly

Line 171: Other genes besides MAPK8IP2 might contribute to PMS such as SULT4A1 which function has been studied in rodent neurons (Culotta et al. JNeurosci 2019).

-We  thank the reviewer for his suggestion and included this paper

Lines 270-273. The citations related to the effect of IGF1 on human neurons are wrongs, the correct one is Shcheglovitov et al. Nature 2013

-We corrected the text accodingly

.

Line 294: Litium was also proposed as a potential drug to treat PMS patients with a specific mechanism described in Darville et al EBioMedicine, 2016. This important finding was not cited.

-We cited the suggested paper

Line 306: Finally other potential therapies with mGlu5 PAM were not cited, Verpelli et al JCB 2011, Vicidomini et al Mol Psychiatry 2017 and Wang et al. Nat Commun 2016.

-We  thank the reviewer for his suggestion and included those  papers

In general, the authors should carefully check the cited literature in support of the written text. Several papers were cited in the wrong context, this couldn’t be accepted for a review.  

Round 2

Reviewer 2 Report

None

Author Response

Thank you.